

# The implementation of rare events logistic regression to predict the distribution of mesophotic hard corals across the main Hawaiian Islands

Lindsay M. Veazey[1], Erik C. Franklin[2], Christopher Kelley[3], John Rooney[4], L. Neil Frazer[5] and Robert J. Toonen[6]

[1] Department of Biology, University of Hawaii at Manoa, Honolulu, HI, United States
[2] School of Ocean and Earth Science and Technology, University of Hawaii, Hawaii Institute of Marine Biology, Kaneohe, HI, United States
[3] The Hawaii Undersea Research Lab, University of Hawaii at Manoa, Honolulu, HI, United States
[4] Joint Institute for Marine and Atmospheric Research, University of Hawaii at Manoa, Honolulu, HI, United States
[5] Department of Geology and Geophysics, University of Hawaii at Manoa, Honolulu, HI, United States
[6] Hawaii Institute of Marine Biology, University of Hawaii at Manoa, Kaneohe, HI, United States

Corresponding author
Lindsay M. Veazey,
lindsayv@Hawaii.edu

## ABSTRACT

Predictive habitat suitability models are powerful tools for cost-effective, statistically robust assessment of the environmental drivers of species distributions. The aim of this study was to develop predictive habitat suitability models for two genera of scleractinian corals (*Leptoseris* and *Montipora*) found within the mesophotic zone across the main Hawaiian Islands. The mesophotic zone (30–180 m) is challenging to reach, and therefore historically understudied, because it falls between the maximum limit of SCUBA divers and the minimum typical working depth of submersible vehicles. Here, we implement a logistic regression with rare events corrections to account for the scarcity of presence observations within the dataset. These corrections reduced the coefficient error and improved overall prediction success (73.6% and 74.3%) for both original regression models. The final models included depth, rugosity, slope, mean current velocity, and wave height as the best environmental covariates for predicting the occurrence of the two genera in the mesophotic zone. Using an objectively selected theta ("presence") threshold, the predicted presence probability values (average of 0.051 for *Leptoseris* and 0.040 for *Montipora*) were translated to spatially-explicit habitat suitability maps of the main Hawaiian Islands at 25 m grid cell resolution. Our maps are the first of their kind to use extant presence and absence data to examine the habitat preferences of these two dominant mesophotic coral genera across Hawaiʻi.

# INTRODUCTION

Consistent and pervasive deterioration of marine ecosystems worldwide highlights significant gaps in current management of ocean resources (*Foley et al.*, *2010*; *Douvere*, *2008*; *Crowder & Norse*, *2008*). One such gap is the data required for informed marine

spatial planning, a management approach that synthesizes information about the location, anthropogenic use, and value of ocean resources to achieve better management practices such as defining marine protected areas and implementing harvesting restrictions (*Jackson, Trebitz & Cottingham*, *2000*; *Larsen et al.*, *2004*). The creation of spatial predictive models for improved marine planning is a relatively low-cost and non-invasive technique for projecting the effects of present-day human activities on the health and geographic distribution of marine ecosystems.

Defining and managing the biological and physical boundaries of ecosystems is a complicated but essential component of marine spatial planning (*McLeod et al.*, *2005*). The heterogeneous nature of ecological datasets can require the time-intensive development of problem-specific ecosystem models (*Cramer et al.*, *2001*; *Tyedmers, Watson & Pauly*, *2005*). Scientists frequently use straightforward, easy-to-implement regression methods to analyze complex datasets. The development of software accessible to relative novices has contributed to the growing popularity of regression methods (e.g., *Lambert et al.*, *2005*; *Tomz, King & Zeng*, *2003*).

Here, we employ a logistic regression with rare events corrections (*King & Zeng*, *2001*) to analyze the presence and absence data of two coral genera (*Leptoseris* and *Montipora*) and, thus, develop a predictive framework for the geographic mapping of mesophotic coral reef ecosystems (MCEs) across the main Hawaiian Islands. Mesophotic coral ecosystems, located at depths of 30–180 m, are considered to be extensions of shallow reefs because they harbor many of the same reef species present at shallower depths, and are also oases of endemism in their own right (*Grigg*, *2006*; *Lesser et al.*, *2010*; *Kane, Kosaki & Wagner*, *2014*; *Hurley et al.*, *2016*). MCE habitats are formed primarily by macroalgae, sponges, and hard corals tolerant of low light levels (*Lesser, Slattery & Leichter*, *2009*). Corals of genus *Montipora* colonize primarily the shallow reef zone (<30 m), but some species, particularly *Montipora capitata* (*Rooney et al.*, *2010*), are able to extend their settlement into mesophotic depths. Corals of genus *Leptoseris* construct extremely efficient, light-capturing skeletons that facilitate their habitation of the lower mesophotic zone (*Kahng et al.*, *2012*) and are considered to be exclusively mesophotic dwellers (*Kahng & Kelley*, *2007*).

Ecological studies in the mesophotic zone are sharply limited in contrast to the shallower photic zone more accessible by open circuit SCUBA, but steady advances in diving, computing, and remotely operated vehicle technologies continue to facilitate interdisciplinary mesophotic research (*Pyle*, *2000*; *Puglise et al.*, *2009*). Mesophotic research in Hawaiʻi has been conducted primarily in the ʻAuʻau Channel, Maui, a relatively shallow, semi-enclosed waterway between the islands of Maui and Lānaʻi that is among the most geographically sheltered and accessible areas in the Hawaiian Archipelago, and, as a result, much of the existing video and photo records of MCEs are from this area. This concentration of historic surveys highlights the importance of creating a pan-Hawaiʻi predictive habitat model to identify likely areas of MCEs across unexplored areas of Hawaiʻi's mesophotic zone. Increasing our knowledge about the habitat preferences of the deep extensions of shallow coral species is critical given that approximately 40% of shallow (<20 m) reef-building corals face a heightened extinction risk from the effects of climate change (*Carpenter et al.*, *2008*). Here, we model the habitat associations of mesophotic

scleractinian corals because of both their intrinsic biological value as well as their potential to recolonize globally threatened shallow reef areas and serve as a refuge to mobile reef organisms (*Bongaerts et al.*, *2010*; *Kahng, Copus & Wagner*, *2014*).

Previous research about the environmental variables driving mesophotic scleractinian colonization in Hawaiʻi suggests that distinct variation in community structure exists between the upper (30–50 m) and mid to lower mesophotic (50–180 m) depths (*Rooney et al.*, *2010*; *Kahng et al.*, *2010*; *Kahng, Copus & Wagner*, *2014*). Potentially influential environmental variables include photosynthetically active radiation (PAR) levels (*Goreau & Goreau*, *1973*; *Fricke, Vareschi & Schlichter*, *1987*; *Kahng & Kelley*, *2007*; *Kahng et al.*, *2010*), isotherms (*Grigg*, *1981*; *Kahng & Kelley*, *2007*; *Rooney et al.*, *2010*), and hard substrate availability (*Kahng & Kelley*, *2007*; *Costa et al.*, *2012*). *Rooney et al.* (*2010*) noted that hard coral abundance declined dramatically below 100 m despite high (>25%) availability of colonizable substrate; this sudden reduction in coral cover occurs at increasingly shallower depths across the northwestern Hawaiian Ridge and may be driven by the synchronously shallower occurrence of isotherms.

Light and temperature intensity (*Jokiel & Coles*, *1977*; *Rogers*, *1990*), physical stress (e.g., wave energy or uncontrolled tourism) (*Dollar*, *1982*; *Nyström, Folke & Moberg*, *2000*; *Franklin, Jokiel & Donahue*, *2013*), and availability of colonizable substrate (*Jokiel et al.*, *2004*; *Franklin, Jokiel & Donahue*, *2013*) are known drivers of shallow (<30 m) reef coral distributions across the world. We expect that our model will capture the influence of these abiotic variables on the distribution of mesophotic corals, especially in the shallower mesophotic zone. We speculate that our model may detect unexpected drivers of *Leptoseris* distribution, particularly because *Leptoseris* is known to colonize deeper depths that bear little resemblance to shallow reefs (*Lesser, Slattery & Leichter*, *2009*; *Rooney et al.*, *2010*). Finally, previous predictive modeling research about the drivers of Hawaiian mesophotic coral colonization identified depth, distance from shore, euphotic depth, and sea surface temperature as potentially influential environmental variables (*Costa et al.*, *2012*; *Costa et al.*, *2015*). Our novel modeling approach utilizes all observational data (corals present and absent), which we believe will offer more insight into the dynamics that facilitate and inhibit coral colonization across the mesophotic zone.

## MATERIALS AND METHODS

### Organismal and environmental data

The Hawaiʻi Undersea Research Laboratory (HURL) and the Pacific Islands Fisheries Science Center (PIFSC) provided video and photo records from MCEs in the Hawaiian Islands for our analyses. This imagery came from 19 dives conducted using submersibles, remotely operated vehicles (ROVs), autonomous underwater vehicles (AUVs), and tethered optical assessment devices (TOADs) in the ʻAuʻau Channel, Maui (13 dives) and two other geographically distinct regions: south Oʻahu (5 dives) and southeast Kauaʻi (1 dive). These dives were conducted between 2001–2013. We analyzed dive video using the Coral Point Count with Excel extensions (CPCe) tool (*Kohler & Gill*, *2006*) in combination with a modified PIFSC 2011 mapping protocol (*Pacific Islands Benthic Habitat Monitoring Center*

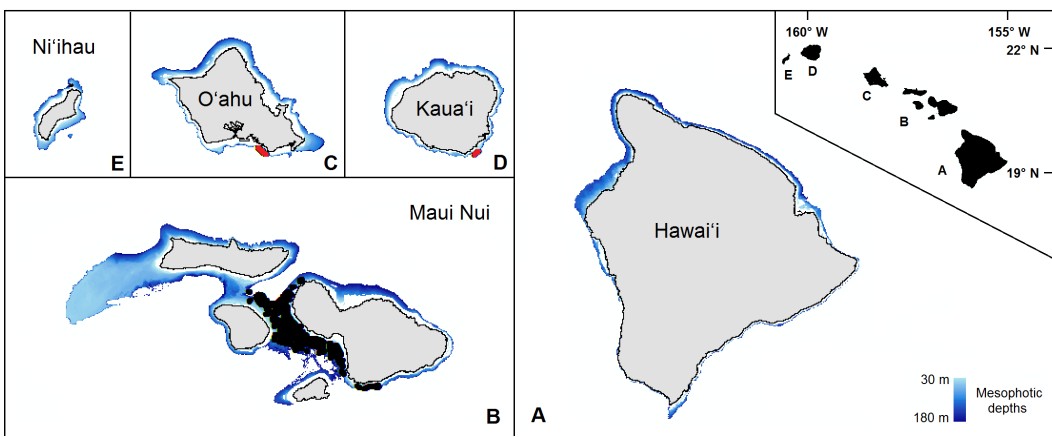

**Figure 1 The mesophotic zone of the main Hawaiian Islands.** The study domain, demarcated in blue, encompasses the mesophotic zone (30–180 m in depth) of the main Hawaiian Islands. Black circles are the observations from the pre-existing Maui Nui dataset. Red circles are the previously unprocessed observations in south Oʻahu and southeast Kauaʻi.

**Table 1 Number of field observations for each coral genus.**

| Source | No. observations | *Leptoseris* | *Montipora* |
|---|---|---|---|
| Oʻahu | 2,645 | 192 | 0 |
| Kauaʻi | 112 | 38 | 3 |
| Maui | 19,957 | 708 | 791 |
| *Total* | 22,714 | 938 | 794 |

(*PIBHMC*), *2015*). PIFSC has used this type of combined analysis, referred to as the random five point overlay method (RFPOM), to process coral reef ecosystem benthic imagery throughout the U.S. Pacific Islands Region since August 2011, and our use of it ensures database consistency with regions processed prior to this study. The CPCe software placed five points randomly on each snapshot, which we then assessed for coral presence. If any of the five points was on coral, that observation was recorded as a "presence." In an effort to evaluate the accuracy of RFPOM, we counted all corals in 200 randomly selected screengrabs and found that this method misses 2.4% of coral observations recorded in these images. We categorized corals by genus, because both *Montipora* (*Forsman et al.*, *2010*) and *Leptoseris* (*Luck et al.*, *2013*) contain species complexes that remain the subject of taxonomic uncertainty which prevent us from being able to reliably identify corals to the species level from photographs.

We recorded snapshots every 30 s for the duration of each dive video. In addition to an existing database of 40,193 records from dives in the ʻAuʻau Channel, 3517 new snapshots were collected from the additional dives across south Oʻahu and Kauaʻi (Fig. 1). Of these 43,710 total images, 20,980 were discarded because either: (1) crucial observational data were absent; (2) they were redundant due to an extended stationary period; or (3) they fell outside the study depth range of 30–180 m. Of the remaining 22,714 records, we analyzed 2,757 unprocessed images using the RFPOM (Table 1).

We selected our environmental covariates, listed in Table 2, based on the sufficiency of the data and the potential significance of each variable as an indicator of hard coral habitat suitability (e.g., *Dolan et al.*, *2008*; *Rooney et al.*, *2010*; *Costa et al.*, *2012*). We defined summer and winter seasons as May–September and October–April, respectively (*Kay*, *1994*; *Rooney et al.*, *2010*). We delineated significant wave height estimates and mean current velocities by season and direction. We extracted and averaged significant wave height data from 144 days per season of twenty-four hour PacIOOS Simulating WAves Nearshore (SWAN) regional wave models estimated values for 2011–2015 (see website: http://oos.soest.Hawaii.edu/las/). Mean current velocity values were available from 0:00–21:00 every three hours for all months from 2013–2015; for each season and direction, 48 mean current velocity values were extracted and averaged from the PacIOOS Regional Ocean Modeling System (see website: http://oos.soest.Hawaii.edu/las/). This model has a 4 km horizontal resolution with 30 vertical levels across seafloor terrain. We sourced monthly MODIS Aqua Chlorophyll *a* averages for the year 2012 from the NOAA PIFSC OceanWatch Live Access Server (see website: http://oceanwatch.pifsc.noaa.gov/). Using the *Morel* (*2007*) method, we applied the following cubic polynomial equation to obtain logged euphotic depth:

$$\log_{10} Z_{eu} = 1.524 - 0.436x - 0.0145x^2 + 0.0186x^3, \tag{1}$$

where $x$ represents the measured Chlorophyll *a* concentrations (mg/m$^3$) at sea surface. Euphotic depth is the depth at which the level of photosynthetically active radiation (PAR), a limiting factor for many heterotrophic mesophotic corals, is at 1% of surface PAR. In total, we used 14 environmental predictor variables to shape our model (Table 2) (Figs. S1–S5).

The spatial resolution of the bathymetry data was 50 m × 50 m for all islands. We resampled the bathymetry raster to a cell size of 25 × 25 m consistent with a conservatively estimated ±25 m positioning error margin observed at a depth of ∼800 m. We estimated an average camera swath value of 3.24 m (range 2.45–4.54 m) based on previous measurements from 19 still image screenshots taken when the submersible was located at different heights above the seafloor. Our geopositional error for the images is ±5 m and we can expect that the location data are within a circle with a 10 m diameter. Our observation sampling area is projected out from the location area a distance of ≤5 m. Addition of a conservative 5 m observation area buffer to the location error area produces an observational data position of ±20 m from the given coordinates of a data point.

We removed all subsampling within cells due to slight variations in camera angles or vessel speed through a point-to-raster conversion. We categorized all cells with ≥1 presence observation as "present" cells and all cells with only absence observations as "absent" cells. This removal of multiple observations within the same 25 × 25 m pixel effectively eliminates pseudoreplication within the data. We used ArcToolbox and the Benthic Terrain Modeler Toolbox to calculate slope, curvature, rugosity, and aspect (compass direction) values (*Wright et al.*, *2012*). We performed a spatial join based on proximity to observation point data to assign values for surface Chlorophyll *a* concentration, mean current velocities, distance from shore, and significant wave heights.

Veazey et al. (2016), *PeerJ*, DOI 10.7717/peerj.2189

**Table 2  List of all variables considered for inclusion in our analyses.**

| Variable | Category | Variable description | Source | Resolution | Variable |
|---|---|---|---|---|---|
| Biological (response) | Hard coral | Presence/absence between 30–180 m in depth | PIFSC, HURL optical validation data | NA | *Leptoseris* |
| | | | | | *Montipora* |
| Environmental (predictor) | Light availability | The depth of the euphotic zone (PAR 1%) determined using the Morel method (*2007*) | NOAA Oceanwatch Live Access Server; NASA, 2014 | 4 km × 4 km | Mean euphotic depth (m) |
| | Topography | Seafloor complexity calculated with the ArcGIS BTM Terrain Ruggedness tool | | | Rugosity (unitless) |
| | | Depth of seafloor | | | Depth (m) |
| | | Rate of change calculated with the ArcGIS BTM Slope tool | | | Slope (degrees) |
| | | Curvature of the seafloor calculated using the ArcGIS Curvature tool | | | Curvature (degrees of degrees) |
| | | Hardness of seafloor detected by acoustic backscatter | USGS, 1998; University of Hawaii SOEST, 2014 | 50 m × 50 m resampled to 25 m × 25 m | Substrate hardness (unitless) |
| | | Distance of observation point to nearest coastline | | | Distance to coastline (m) |
| | | Compass direction of maximum slope calculated using the ArcGIS Aspect tool | | | Aspect (degrees) |
| | Waves/currents | Mean current velocity data obtained per season (winter/summer) for depths: 200, 150, 125, 100, 75, 50, 30 m | PacIOOS Hawaii Regional Ocean Model | 4 km × 4 km | Mean cur. vel. (northward/summer) (m s$^{-1}$) |
| | | | | | Mean cur. vel. (northward/winter) (m s$^{-1}$) |
| | | | | | Mean cur. vel. (eastward/summer) (m s$^{-1}$) |
| | | | | | Mean cur. vel. (eastward/winter)(m s$^{-1}$) |
| | | Sea surface mean significant wave height | PacIOOS Hawaii SWAN Wave Model | 0.5 km × 0.5 km and 1 km × 1 km | Sig. wave height (summer) (m) |
| | | | | | Sig. wave height (winter) (m) |

## Regression methods

In describing the relationship between a response variable and one or more predictor variables, we use a logistic regression model because the response variable is dichotomous (*Hosmer & Lemeshow*, *2004*). The ordinary logistic regression (OLR) model is defined as:

$$\theta = \text{expit}(\mu) = \frac{1}{1 + \exp(-\mu)}, \tag{2}$$

where $\theta$ is the probability that the species of interest is present ($y = 1$), and $1 - \theta$ is the probability it is absent ($y = 0$). The logit function is the inverse of the expit function, and

$$\text{logit}(\theta) = \mu = \beta_0 + \beta_1 x_1 + \cdots + \beta_n x_n \tag{3}$$

is the linear sum of predictor variables, $x_1$, $x_2$, ..., $x_n$, with intercept $\beta_0$ andregression coefficients $\beta_1$, $\beta_2$, ..., $\beta_n$. In the language of generalized linear models (GLM), OLR is said to have the logit function as its link function and the expit function as its inverse link function. Logistic regression provides a straightforward, meaningful interpretation of the relationship between a dichotomous dependent variable $y$ and a set of predictor variables (*Allison*, *2001*).

Despite the popularity of OLR, it may yield extremely biased results when an imbalance exists in the proportion of the response variable data (e.g., such as in our case, when $y = 0 \gg y = 1$) (*Van Den Eeckhaut et al.*, *2006*). *King & Zeng* (*2001*) coined the term "rare events logistic regression" to describe their corrective methodology in dealing with unbalanced binary event data:

1. The first step requires the selection of a representative sample. Though researchers generally prefer to work with more uniform response data (e.g., *Liu et al.*, *2005*), selection of an unusually high proportion of the rare event (in this case, $y = 1$) to "balance" the dataset and increase $\theta$ estimates will yield nonsensical results. We divided the data in half to create our training and testing datasets and checked that each set of observations had an approximately equal proportion ($\overline{y}$) of presence observations to better reflect the "true state" of the full dataset.

2. The second step rectifies any bias that might be introduced when dividing the dataset. This prior correction on the intercept ($\beta_0$) can be calculated as:

$$\hat{\beta}_0 = \tilde{\beta}_0 - \ln\left[\left(\frac{1-\tau}{\tau}\right)\left(\frac{\overline{y}}{1-\overline{y}}\right)\right]; \tag{4}$$

here, $\hat{\beta}_0$ is the corrected intercept, $\tilde{\beta}_0$ is the uncorrected intercept, $\tau$ is the true proportion of 1s in the population; and $\overline{y}$ is the observed proportion of 1s in the training sample.

3. The third step rectifies any underestimation of the probabilities of the independent variables $\beta_{1...n}$ from the substitution of the intercept value, obtained as:

$$P(y_i = 1) = \tilde{\theta}_i + C_i, \tag{5}$$

where the correction factor $C_i$ is given by:

$$C_i = (0.5 - \tilde{\theta}_i)\tilde{\theta}_i(1 - \tilde{\theta}_i)XV(\tilde{\beta}_i)X', \tag{6}$$

where $X$ is a $1 \times (n+1)$ vector of values for each independent variable $\beta_i$, $X'$ is the transpose of $X$, and $V(\tilde{\beta}_i)$ is the variance covariance matrix. We obtained the improved probability estimates through estimation of $\beta_i$ via $\tilde{\beta}_i$, thereby considered "mostly" Bayesian (*King & Zeng*, *2001*). Our priors in this case would be uninformative, which means that we lack sufficient knowledge to estimate the probability distributions of our data and our parameter of interest, $\theta$. This is often the case when working with sparse ecological datasets. As the uninformative prior for a regression coefficient with domain $(\infty, -\infty)$ is uniform, a full Bayesian estimation with uninformative priors is equivalent to a traditional logistic regression. Therefore, this correction is effectively a correction to the approximate Bayesian estimator, and its addition improves our regression by lowering the mean squared error of our estimates. We implemented this rare events logistic regression using the 'Zelig' package run in R (*Imai, King & Lau*, *2008*; *Choirat et al.*, *2015*).

We constructed a correlation scatterplot matrix per coral genus to observe correlation levels between all variables. In choosing which highly correlated variables to exclude from the analyses, we followed the criteria outlined by *Dancey & Reidy* (*2004*) and *Tabachnick & Fidell* (*1996*), who suggest a cutoff correlation value of 0.7. Only mean significant wave height parsed by season consistently overreached this threshold; the covariate that was least correlated with the response variable was removed. We excluded predictors that lacked a clear distribution pattern or correlated minimally ($<0.05$) with the response variable.

One of the more studied habitat preferences of *Leptoseris* and *Montipora* is the influence of depth on their distribution (*Rooney et al.*, *2010*; *Costa et al.*, *2012*; *Kahng et al.*, *2010*). Increasing depths often correlate with greater distance from shore. The inclusion of squared terms (e.g., $x_2 = x_1^2$) in our regression equation $\text{expit}(\theta) = \beta_0 + \beta_1 x_1 + \cdots + \beta_n x_n$ permits the logistic curve to reflect the bell curve shape expected in plotting the distribution of these animals across a range of depths or distance from shore. In order to account for these trends, we added Depth Squared and Distance Squared as potential variables for consideration in our final model. As depth or distance increases, its square increases even more rapidly, allowing the squared term to eventually dominate and "pull down" the probability curve.

We withheld 50% of our information per genus as testing (i.e., validation) data. Using the remaining 50% (our training data), we performed the rare events corrected logistic regression described above. Using an exhaustive iterative algorithm (*Calcagno, Mazancourt & Claire*, *2010*), we modeled all possible combinations of included covariates. We ranked models using the corrected Akaike information criterion (AICc) (*Hurvich & Tsai*, *1989*), which is considered an excellent comparative measurement of model strength, especially for sparse datasets. For both genera, the models with the lowest (lowest = best) AICc scores were lower than the "second best" AICc scores by at least 2 (i.e., $\Delta$ AICc $\geq 2$), indicating strong preference for the best model (e.g., (*Hayward et al.*, *2007*)).

In an ideal and unrealistic study, all biotic and abiotic components of a model would be homogenous and evenly distributed across a sampling space. Our sampling design includes overlapping submarine dive tracks and the inherent heterogeneity of the marine environment, which could problematically violate our model's underlying assumption

**Table 3 Summary statistics for theoretical semivariogram models.**

| Genus | Sum of squares | Input $\sigma^2$ | Input $\psi$ | Actual $\sigma^2$ | Actual $\psi$ | Actual $\tau^2$ |
|---|---|---|---|---|---|---|
| *Leptoseris* | 2940.671 | 0.055 | 218 | 0.051 | 206.909 | 0 |
| *Montipora* | 14013.610 | 0.020 | 390 | 0.032 | 390.000 | 0.003 |

regarding the independence of our biological and environmental data. We removed all instances of pseudoreplication (multiple observations in one grid cell) when we assigned each grid cell to a category of "corals present" or "corals absent." After we removed subsampling within our observational data, we checked for the presence of clustering, or spatial autocorrelation, within these data. Uncorrected spatial autocorrelation between observational data points confounds and undermines any biological inferences drawn from model predictions.

We checked small-scale, local spatial autocorrelation using Geary's C statistic (*Geary*, *1954*), based on the deviations in the responses of observation points with one another:

$$C = \frac{n-1}{2S_0} \frac{\sum_i \sum_j w_{ij}(x_i - x_j)^2}{\sum_i (x_i - \overline{x})^2}. \tag{7}$$

Here, $x$ is the variable of interest, $i$ and $j$ are locations (where $i \neq j$), $w_{ij}$ represents the components of the weight matrix, and $S_0$ is the sum of the components of the weight matrix. Geary's C ranges from 0 (maximal positive autocorrelation) to 2 for high negative autocorrelation. In the absence of autocorrelation, its expectation is 1 (*Sokal & Oden*, *1978*).

We also examined global spatial autocorrelation using Moran's I statistic, which measures cross-products of deviations from the mean (*Moran*, *1950*):

$$I = \frac{n}{S_0} \frac{\sum_i \sum_j w_{ij}(x_i - \overline{x})(x_j - \overline{x})}{\sum_i (x_i - \overline{x})^2}. \tag{8}$$

Moran's I values generally range from −1 to 1, with 0 as the expectation when no spatial autocorrelation is present.

We also verified the spatial independence of our observational point data using a semivariogram, which is a graphical method of quantifying spatial correlation in a set of points (Figs. 2–3). We selected our theoretical semivariogram to fit the empirical semivariance using the ordinary least squares (OLS) method (*Jian, Olea & Yu*, *1996*; *Kendall et al.*, *2005*). The spherical model had the best quantitative fit based on OLS estimates (Table 3). For each dataset, the low thresholds at which semivariance stopped increasing indicated the almost complete absence of spatial autocorrelation for each genus.

## Model assessment

Evaluation of the rare events logistic regression model output is more complicated than for the typical linear model. For example, $R^2$ values, although calculated, have little

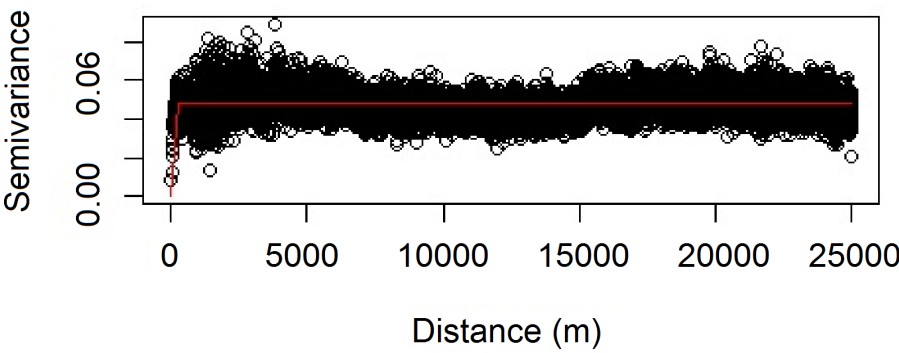

**Figure 2** Modeled spherical semivariogram for *Leptoseris.*

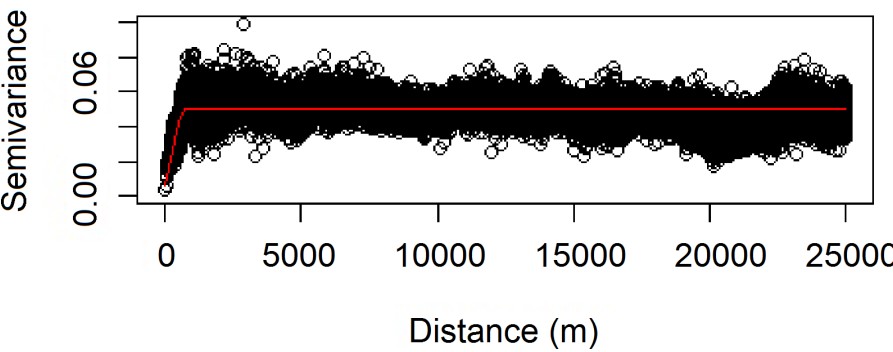

**Figure 3** Modeled spherical semivariogram for *Montipora.*

applicability to logistic regressions and are therefore ignored (*Menard, 2000; Peng, Lee & Ingersoll, 2002*). Sample size and selected threshold largely influence the results of the Hosmer and Lemeshow goodness-of-fit test (*Hosmer et al., 1997*). Accordingly, we use model classification accuracy as a second measure of goodness-of-fit (in addition to $\Delta$AICc). We want to maximize true positives (TP) and true negatives (TN) while minimizing false positives (FP) and false negatives (FN). The sensitivity-specificity sum maximization approach (*Cantor et al., 1999*) therefore maximizes

$$SS_{\max} = \frac{TP}{TP+FN} + \frac{TN}{TN+FP}, \tag{9}$$

which is equivalent to finding the point on the ROC (receiver operating characteristics) curve at which the tangent slope is 1, indicating the optimal cutoff point at which "cost" (here, the number of FN and FP) and "benefit" (the number of TN and TP) is balanced. We chose this technique because we aim to identify regions devoid of hard corals as well as regions deemed potentially suitable for habitation.

ROC curves plot the true positive test rate against the false positive test rate across different theta cutoff points (*Hadley & McNeil, 1982*). We calculated values for sensitivity and specificity for threshold increments of $0.005 \pm 1$ standard deviation of the rounded mean for each model. Because each theta threshold value varied based on the genus and model, the threshold-independent area under the curve (AUC) test statistic best reflects the predictive accuracy of the model.

In addition to creating ROC curves, we also took into account the overall prediction success of each model, given as:

$$OPS = \frac{TP + TN}{TP + TN + FP + FN}. \tag{10}$$

Overall prediction success is a measure of total correct classification of both present and absent observations. While this is a good final assessment of model classification error, consideration of the prediction success alone is not a viable evaluation method when binary data is highly imbalanced, as a value given by this method may primarily represent model success in identifying the most common observation type (*Fielding & Bell, 1997*). We plotted our sensitivity and specificity values on a ROC curve to show how each model performed relative to chance (Fig. 4). All models fall in the range $0.7 \leq AUC < 0.9$, which indicates good discrimination and reliability of model predictions (*Hosmer & Lemeshow, 2004*).

We also created maps of individual and summed predicted occurrence probabilities of both coral genera across the main Hawaiian Islands and ran a hotspot analysis using the ArcGIS Getis-Ord $G_i^*$ Hotspot Analysis tool. We constructed a polygon fishnet composed of $1 \times 1$ km cells which encompassed all islands. We summed each $25 \times 25$ m raster cell value for probability of *Leptoseris* occurrence and probability of *Montipora* occurrence. We performed a spatial join of raster cell values within each polygon for an average value of summed probabilities. The Getis-Ord $G_i^*$ statistic identifies clusters within these polygons that display values higher in magnitude than random chance would permit. The Getis-Ord local statistic is given as:

$$G_i^* = \frac{\sum_{j=1}^{n} w_{i,j} x_j - \overline{X} \sum_{j=1}^{n} w_{i,j}}{S \sqrt{\frac{1}{n-1} \left[ n \sum_{j=1}^{n} w_{i,j}^2 - \left( \sum_{j=1}^{n} w_{i,j} \right)^2 \right]}}. \tag{11}$$

Here, $w_{i,j}$ represents the spatial weights between features $i$ and $j$; $n$ represents the total number of features; $x_j$ is the attribute value for feature $j$; $\overline{X} = \frac{1}{n} \sum_{j=1}^{n} x_j$; and

$$S = \sqrt{\frac{1}{n} \sum_{j=1}^{n} x_j^2 - (\overline{X})^2}.$$
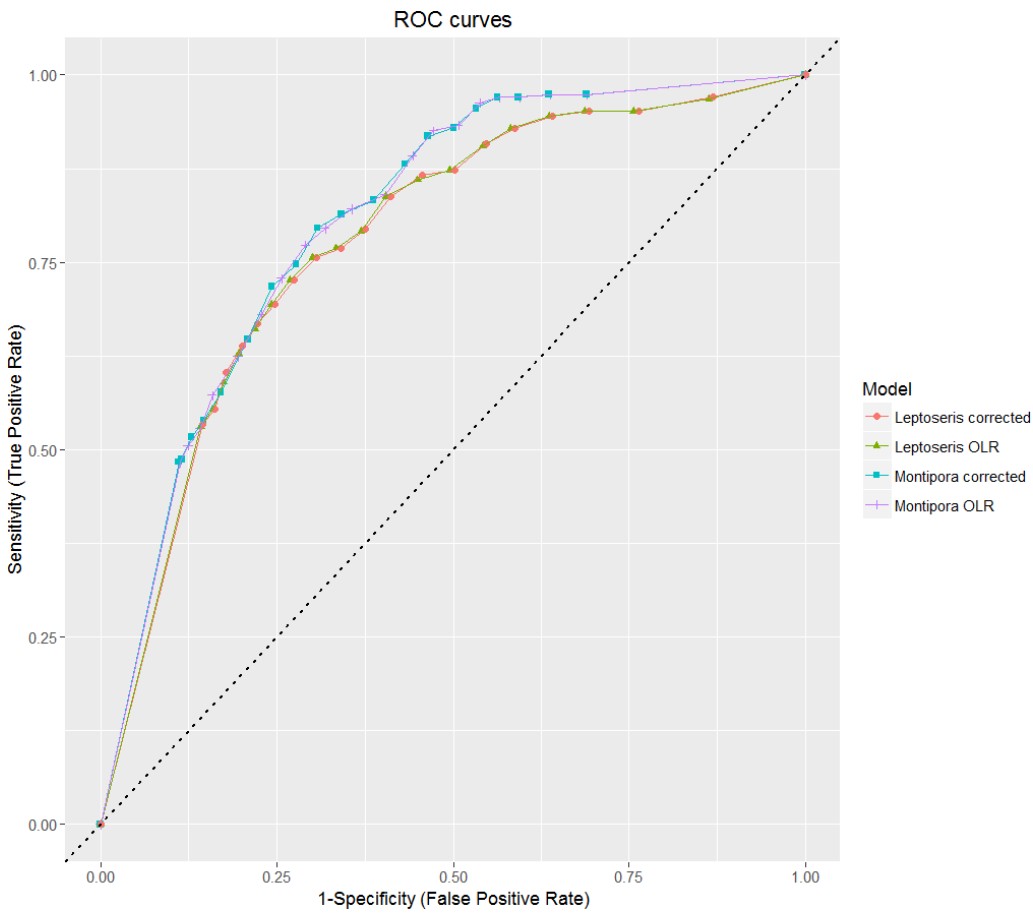

**Figure 4  ROC curves for all models.** AUC values for all models fall in between 0.7 and 0.9, which indicates predictive reliability. The dashed line from (0, 0) to (1, 1) indicates the null threshold at which model performance is considered unacceptable (<0.5).

## RESULTS

Geary's $C$ test statistic is a measure of local (small-scale) spatial autocorrelation; in the absence of correlation, 1 is the expected value of Geary's C. Moran's I is a measure of global (large-scale) spatial autocorrelation; in the absence of correlation, a value of 0 is expected for the Moran's $I$ test statistic. For our *Leptoseris* dataset, Geary's $C = 0.990$; for our *Montipora* dataset, Geary's $C = 0.996$. For our *Leptoseris* dataset, Moran's $I = 0.006$; for our *Montipora* dataset, Moran's $I = 0.003$. These values do not indicate any local clustering or global spatial autocorrelation within either dataset. We observed negligible levels of autocorrelation up to ∼100 m for *Montipora* (Fig. 3). By ensuring that spatial autocorrelation is not present in our data, we do not violate the assumption that our response data are independently observed, which enables us to draw robust conclusions about the ecological factors influencing the distribution of these coral genera within the mesophotic zone across the main Hawaiian Islands.

The OLR covariate coefficients were modified using the rare events corrections proposed by *King & Zeng* (*2001*), resulting in a change in predictive power (Table 4). Rare events

**Table 4   Predictive models output.**  Results by genus: theta threshold subscripts indicate model type and training and validation (c-v) outputs. Sensitivity and specificity totals apply to training data only.

| Genus | $\theta$ threshold | TP | TN | FP | FN | $TP_{c-v}$ | $TN_{c-v}$ | $FP_{c-v}$ | $FN_{c-v}$ | Sensitivity | Specificity | $SS_{max}$ | OPS | $OPS_{c-v}$ | AUC |
|---|---|---|---|---|---|---|---|---|---|---|---|---|---|---|---|
| *Leptoseris* | $\theta_{OLR} =$ 0.065 | 223 | 4,133 | 1,525 | 84 | 219 | 3,757 | 1,894 | 94 | 0.7264 | 0.7305 | 1.4569 | 73.0% | 66.7% | 0.782 |
| | $\theta_{corr} =$ 0.067 | 220 | 4,168 | 1,490 | 87 | 182 | 4,000 | 1,651 | 131 | 0.7166 | 0.7367 | 1.4533 | 73.6% | 70.1% | 0.780 |
| *Montipora* | $\theta_{OLR} =$ 0.064 | 200 | 4,299 | 1,536 | 69 | 159 | 4,453 | 1,406 | 86 | 0.7435 | 0.7368 | 1.4803 | 73.7% | 75.6% | 0.808 |
| | $\theta_{corr} =$ 0.0625 | 198 | 4,336 | 1,499 | 71 | 165 | 4,406 | 1,453 | 80 | 0.7361 | 0.7465 | 1.4792 | 74.3% | 74.9% | 0.809 |

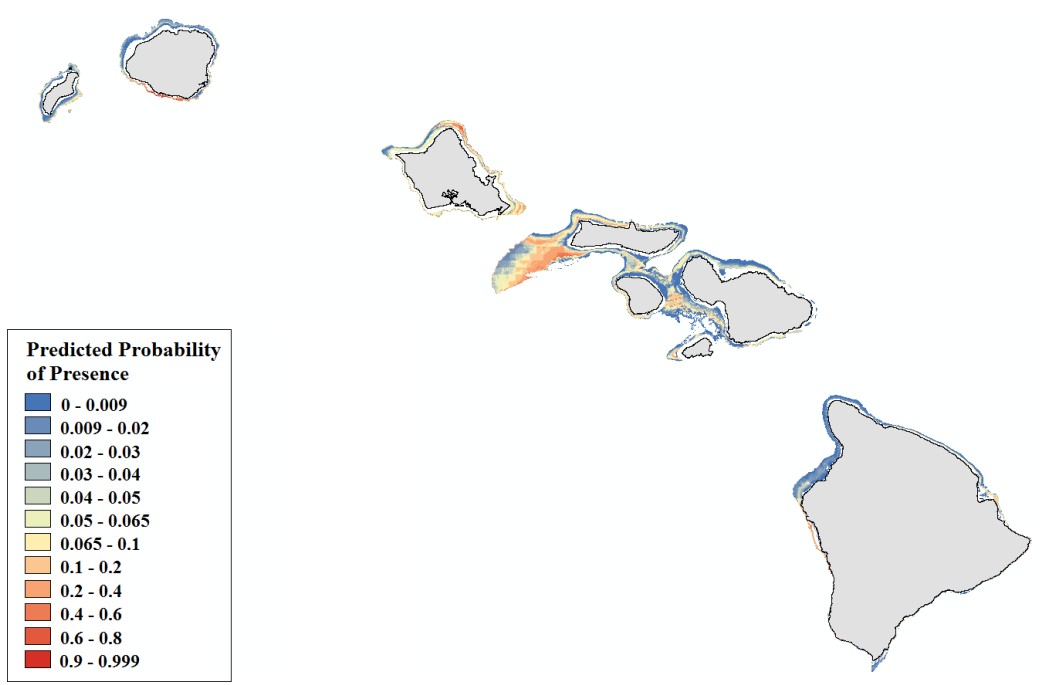

**Figure 5  Modeled area of suitable habitat for *Leptoseris*.** Probability of presence is depicted along a color gradient ranging from red (1; most suitable) to blue (0; least suitable).

corrected models usually performed better than the uncorrected models, in terms of improved specificity and prediction success. Our sensitivity values for both corrected models were slightly lower than the corresponding OLR sensitivities, but in each case, specificity and prediction success values were improved. Additionally, standard errors of the coefficient estimates were lower for corrected models than for uncorrected models (Tables S1–S4).

*Leptoseris* corals inhabit mesophotic regions with high slope and rugosity values, high to moderate perennial current flow, and their occurrence peaks around 100 m (Table S3, Figs. S6–S10). *Montipora* corals peak in occurrence around 60 m and colonize regions less exposed to high energy winter swells (Table S4, Figs. S11–S12). Predicted presence probability values ($\theta$) averaged 0.051 for *Leptoseris* and 0.040 for *Montipora* models in the validation data (Figs. 5–6). These values agree well with the actual presence frequencies in that data (0.052, 0.042). To better interpret these realistically low theta values, we chose a theta threshold to transform the probability estimates to presence/absence values. This is standard practice when examining the results of a rare events logistic regression, but less common when performing OLR (*Liu et al.*, *2005*). Objective selection of a theta threshold on a per-model basis is more scientifically sound than, for example, an arbitrary assignment of 0.5 (*Cramer*, *2003*). The transformed model is valid if a threshold value yields a high percentage of correctly classified observations and a low number of FP and FN observations (*Gobin, Campling & Feyen*, *2001*). We selected an appropriate threshold for each model (Table 4) in order to maximize $SS_{max}$ (*Liu et al.*, *2005*).

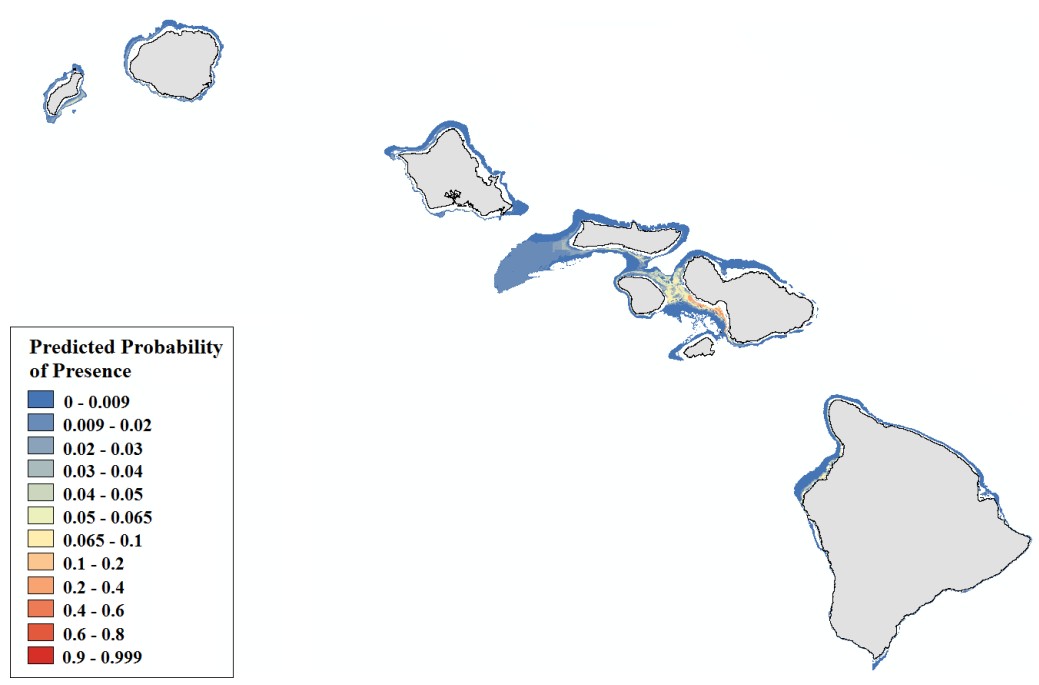

**Figure 6** **Modeled area of suitable habitat for *Montipora*.** Probability of presence is depicted along a color gradient ranging from red (1; most suitable) to blue (0; least suitable).

Our final hotspot maps show the results of our analysis for *Leptoseris, Montipora,* and both genera combined across all islands (Figs. 7–9). We show hotspots of habitat suitability for both coral genera in red for areas of highest suitability and blue for areas of lowest suitability. We identify a cell as a hotspot when the sum of its value and the values of its nearest neighbors is much higher or lower than the mean over all cells. When the local sum of a cluster is very different from the expected value, a statistically significant hotspot is identified ($G_i^*$ statistic $\geq 1.96$ or $G_i^*$ statistic $\leq -1.96$). Neither genus clearly dominated the summed probabilities hotspot identification across any of the islands. Large *Leptoseris* hotspots were identified in southwest Molokaʻi, northeast Oʻahu, west Hawaiʻi, and the central ʻAuʻau Channel. *Montipora* hotspots were identified in east Niʻihau, southwest Kauaʻi, west and south Oʻahu, west Hawaiʻi, and the central ʻAuʻau Channel.

## DISCUSSION

In this study, we used logistic regression with rare events corrections to predict the habitat preferences of two dominant scleractinian coral genera across the entire mesophotic zone surrounding the main Hawaiian Islands. The habitat preferences of *Montipora* in the mesophotic zone appear distinct from those of *Leptoseris*. *Montipora* prefers the middle mesophotic zone (50–80 m) of reefs less exposed to high-energy winter swells. *Leptoseris* prefers steep, rugose slopes and the lower mesophotic zone (>80 m) in regions of high year–round current flow.

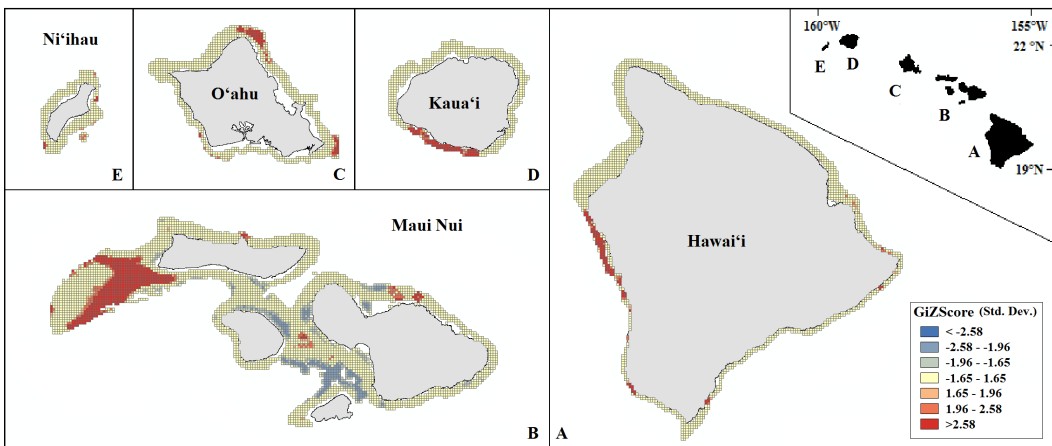

**Figure 7** **Mapped result of our Getis-Ord $G_i^*$ hotspot analysis performed for probability estimates of *Leptoseris* occurrence.** A significant hotspot is $<-1.96$ or $>1.96$; here, all hotspots are shown in red ($>1.96$) or blue ($<-1.96$).

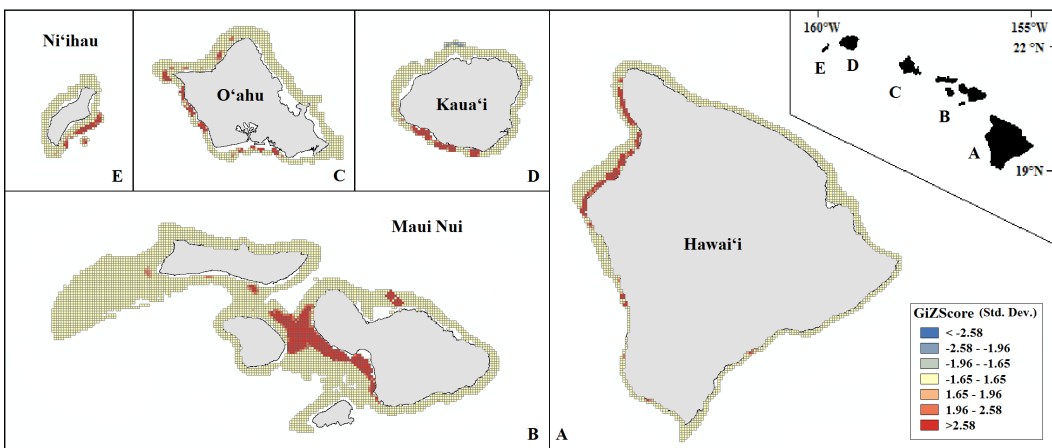

**Figure 8** **Mapped result of our Getis-Ord $G_i^*$ hotspot analysis performed for probability estimates of *Montipora* occurrence.** A significant hotspot is $<-1.96$ or $>1.96$; here, all hotspots are shown in red ($>1.96$) or blue ($<-1.96$).

## Important environmental covariates

Predicted *Montipora* presence peaks at about 60 m (median occurrence probability $= 7.5\%$); *Leptoseris* presence peaks at about 100 m (median occurrence probability $= 7.5\%$). These predictions are consistent with the inferences of *Rooney et al.* (*2010*), which separates mesophotic reefs into three distinct depth sections: upper (30–50 m), branching/plate dominated (50–80 m), and *Leptoseris* dominated ($\geq 80$ m). The depth at which suitability peaks for *Leptoseris* occurs at a range where steep ridges and drop-offs are plentiful in our study region, and therefore the mean preferred depth may be prone to slight overestimation.

In addition to depth, four environmental covariates appeared to influence the distribution of *Leptoseris*: rugosity, slope, summer mean current velocity (northward), and winter mean current velocity (eastward). Scleractinians easily colonize environments

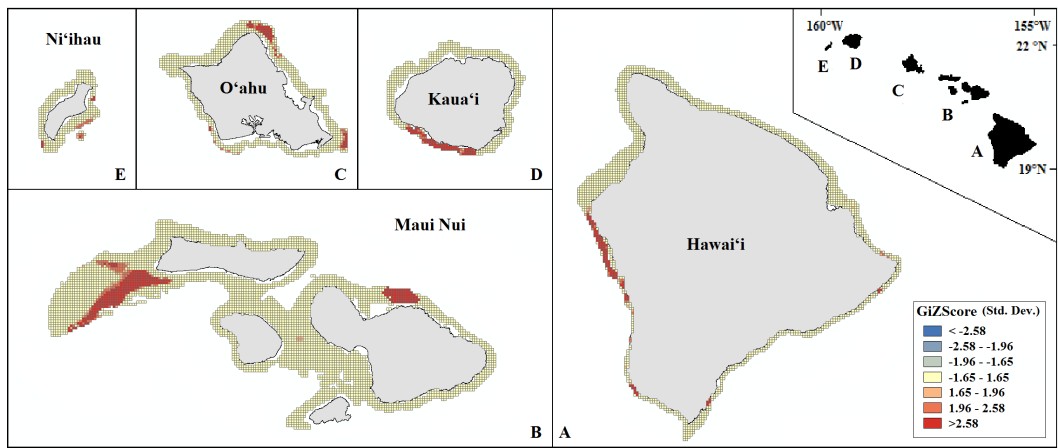

**Figure 9** **Mapped result of our Getis-Ord $G_i^*$ hotspot analysis performed for summed probability estimates of *Leptoseris* and *Montipora* occurrence.** A significant hotspot is $<-1.96$ or $>1.96$; here, all hotspots are shown in red ($>1.96$).

that are relatively calm and rugose due to the larger amount of available surface area, and this positive correlation was reflected in our model. *Leptoseris* habitat preference was also positively associated with slope, which was not observed for *Montipora*. Corals that inhabit the upper mesophotic zone may be more susceptible to damage from debris displaced by high wave energy, and are therefore less likely to colonize steep slopes (e.g., *Harmelin-Vivien & Laboute*, *1986*; *Bridge & Guinotte*, *2013*). The deeper distribution of *Leptoseris* may protect it from damage related to wave intensity, allowing it to colonize slopes (e.g., *White et al.*, *2013*). Another possibility is that the model is picking up drop-offs from masses accreted during the last glacial maximum. These steep drop-offs are present between 90–120 m in the *Leptoseris*-dominated lower mesophotic zone (*Yokoyama et al.*, *2001*; *Webster et al.*, *2004*).

*Leptoseris* also favors well-flushed areas exposed to year-round moderate current flow (i.e., up to 0.3 m/s). The plate-like morphology of *Leptoseris* corals effectively boosts sunlight capture by its symbiotic zooxanthellae and zooplankton capture by the corals themselves, but it also makes the coral vulnerable to smothering by sediment accumulation (*Bak, Nieuwland & Meesters*, *2005*; *Bongaerts et al.*, *2010*; *Marcelino et al.*, *2013*). The success of *Leptoseris* corals in areas of moderate current flow may be related to the improbability of sediment settlement and accumulation. While the model did not capture the same effect of current flow on *Montipora* distribution, we recognize that the morphology of some *Montipora* species is extremely similar to that of *Leptoseris*. We do not expect either genus to readily colonize highly turbid regions, especially given that certain species of heterotrophic *Montipora* are thought to exploit strong currents to meet their energy requirements (*Grottoli, Rodrigues & Palardy*, *2006*; *Rooney et al.*, *2010*).

Substrate hardness, a variable known to influence coral colonization, was notably absent from each model. Substrate hardness values were derived from acoustic backscatter imagery readings. The base resolution of these readings (50 m × 50 m) was not sufficiently detailed for purposes of this analysis. We noted plentiful coral colonization along larger surfaces like lava fingers, the hardness of which would be detectable by backscatter surveys, as

well as across small rock fragments strewn across a sand flat, which would be obscured by the softness of the surrounding benthos. We can conclude that measurements of benthic hardness are not detailed enough for predictive modeling purposes at a 25 × 25 m resolution.

We emphasize that the purpose of this study was to build a pan-Hawai'i predictive habitat map for two dominant coral genera within the mesophotic zone. Because the scope of this study included all main Hawaiian Islands, we were constrained by the coarseness of available full-coverage environmental data. As we build on this analysis, we plan to use our maps to identify areas of interest for further study at higher resolution and to include additional variables currently only available in certain regions, such as light intensity and temperature at depth. For example, our predictive and hotspot maps identify Penguin Bank (southwest Moloka'i) as particularly suitable for *Leptoseris* colonization, which has not been verified by video or photo records. High resolution backscatter data (1 × 1 m) exist for this region, and incorporation of these data into new analyses of subsets of our study area may refine our conclusions.

### Error sources and model reliability

We examined two types of error (false negatives and false positives) and analyzed our models without giving preference to either one. This approach is widely accepted as the best method of overall error minimization (e.g., *Liu et al.*, *2005*; *Fielding & Bell*, *1997*). Rare events corrected models for both *Leptoseris* and *Montipora* achieved levels of specificity and sensitivity well above the null, indicating good predictive power. Additionally, both models attained about 74% overall prediction success. We assumed coral detectability was constant across the study region and that we can therefore consider the true absence observations to be reliable indicators of a potentially unsuitable habitat for corals. For each genus, the model tended to slightly over-predict presence observations; large numbers of false positives lowered sensitivity values. This is inevitable in the analysis of severely imbalanced or sparse binary data; the ongoing addition of presence observations to the dataset will improve overall model classification accuracy.

While the consistent identification of southern coastal areas as suitable is reliable, the comparatively infrequent selection of northern coasts is likely due to the source of the model-building observations. The vast majority of mesophotic exploration has been along southern coastlines, which is often where waters are calmest in Hawai'i. It is speculated that because mesophotic corals are more shielded from winter long-period wave energy than their shallow water counterparts, they are able to flourish at depth along northern coastlines (*Grigg*, *1998*; *Rooney et al.*, *2010*). The addition of data sourced from northern expeditions would likely improve predictive power of the model across north-facing coastlines (*Alin*, *2010*).

We acknowledge that the original data were not collected in a standardized fashion (e.g., variation in vessel traveling speed or differences in data collection vessel and/or quality). Our careful exclusion of overlapping observation points within each 25 × 25 m rectified this sampling design flaw as much as possible and eliminated pseudoreplication.

## Distinctions between coral genera

Our *Montipora* model was simpler than the *Leptoseris* model in that the only variable included other than depth was winter significant wave height. Though uncertainty was highest at lower values of significant wave height, *Montipora* demonstrated a preference in colonizing habitats that experience lower significant wave height during winter. This preference contrasts with *Montipora* species in shallow waters that were more likely to be observed in higher wave height environments (*Franklin, Jokiel & Donahue*, *2013*). This likely influenced the inability of the model to identify any suitable habitat around Niʻihau, where the average winter significant wave height equaled 1.78 m, almost double the mean significant wave height of our model training data (0.91 m). Though mesophotic corals are generally thought to be exempt from the growth limitations faced by shallow water corals in regions of high wave energy, prolonged wave intensity has been shown to negatively affect the colonization of upper mesophotic scleractinians, especially in sloping areas prone to debris avalanches (*Bridge & Guinotte*, *2013*; *Kahng, Copus & Wagner*, *2014*). Continuation of this work might include a more in-depth examination of the relationship of this coral genus with the combined effects of slope of available substrate and exposure to wave energy.

We found no records of *Montipora* presence when processing our Oʻahu dataset, which probably contributed to the very low predicted mean probability of *Montipora* occurrence there (0.1%). We believe this is due in part to the sampling pattern across south Oʻahu; we recorded 62.3% of all observations processed for this region at a depth of 75 m or greater. *Montipora* prevalence is greater in the upper-to-middle mesophotic zone, and the relative deepness of the Oʻahu dives likely influenced their nonappearance in this portion of the dataset. We emphasize that the dearth of *Montipora* observations around Oʻahu is an artifact of the dataset we used to construct our model; *Montipora* corals have been observed in mesophotic depths across Oʻahu (e.g., Fig. 4B, *Rooney et al.*, *2010*). The mean significant wave height across the mesophotic zone was lower across the southern and western coasts (1.50 m) than that observed across the northern and eastern coasts (2.37 m) of the island. As at Niʻihau, we assume that this high northern and eastern average height, coupled with the absence of *Montipora* presences in Oʻahu in the training dataset, greatly impacted our model's ability to detect areas of suitable habitat around the island. The results of our Getis–Ord $G_i^*$ Hotspot Analysis corroborate the findings of *Costa et al.* (*2015*), who used Maximum Entropy software to predict the highest occurrence probability of *Leptoseris* and *Montipora* in the middle and mid-coastal ʻAuʻau Channel, respectively (*Costa et al.*, *2015*).

The factors influencing the distribution of coral species in shallow and mesophotic habitats differ. One of the fundamental drivers of the occurrence and abundance of coral species on shallow reefs in Hawaiian waters is wave stress (*Dollar*, *1982*; *Grigg*, *1983*; *Franklin, Jokiel & Donahue*, *2013*). Given the depth range of MCEs, wave stress is unlikely to serve as a direct influence on coral occurrence but may provide secondary effects as wave events lead to debris reaching MCEs (*Kahng, Copus & Wagner*, *2014*). Furthermore, the decoupled effects of environmental drivers on shallow and mesophotic zones extend between the islands. In shallow reef communities *Montipora* species become relatively more dominant from Hawaii Island to Niʻihau (*Franklin, Jokiel & Donahue*, *2013*), but appear to peak in occurrence in the mesophotic zone of Maui Nui. While strong environmental

drivers influence the distributions of shallow corals, the occurrence patterns of mesophotic corals may reflect a more stable environment with an increased influence of biotic factors such as interspecific competition in a habitat zone with limited light and space resources available.

## CONCLUSIONS

We implemented a rare events corrected logistic regression to determine the most influential environmental predictors of *Montipora* and *Leptoseris* colonization in the mesophotic zone. Habitat preference differences between these genera appear distinct and multi-faceted. *Montipora* thrives in the middle mesophotic zone in areas sheltered from high intensity winter swells, while *Leptoseris* tends to colonize steep, rugose, well-flushed areas in the lower mesophotic zone. Improved understanding of the distribution of mesophotic corals will enable resource managers to propose the construction of seafloor power cables and other offshore infrastructure in areas less likely to contain coral communities. Results will likewise facilitate efforts to protect these communities by supplementing scientific dive planning and strategies for conservation, such as marine spatial planning.

## ACKNOWLEDGEMENTS

Authors thank the many staff members of HURL, PIFSC, and SOEST who helped acquire and process our data. L.V. thanks Dr. David Hondula (Researcher at Arizona State University) for his dedicated GIS instruction. We dedicate this manuscript to the memory of our coauthor, Dr. John Rooney. *A hui hou*, John. This manuscript is HIMB contribution #1662 and SOEST contribution #1947.

### Funding

Funding for this research was provided by the NOAA Coral Reef Conservation Program grant no. NA14NOS4820092. The funders had no role in study design, data collection and analysis, decision to publish, or preparation of the manuscript.

### Grant Disclosures

The following grant information was disclosed by the authors:
NOAA Coral Reef Conservation Program: NA14NOS4820092.

### Competing Interests

Robert Toonen is as an Academic Editor for PeerJ.

### Author Contributions

- Lindsay M. Veazey conceived and designed the experiments, performed the experiments, analyzed the data, contributed reagents/materials/analysis tools, wrote the paper, prepared figures and/or tables, reviewed drafts of the paper.

- Erik C. Franklin conceived and designed the experiments, analyzed the data, contributed reagents/materials/analysis tools, reviewed drafts of the paper.
- Christopher Kelley, John Rooney and Robert J. Toonen contributed reagents/materials/analysis tools, reviewed drafts of the paper.
- L. Neil Frazer conceived and designed the experiments, contributed reagents/materials/analysis tools, reviewed drafts of the paper.

## Data Deposition

The data sets and raw code are included in the Supplemental Information.

## Supplemental Information

Supplemental information for this article can be found online at http://dx.doi.org/10.7717/peerj.2189#supplemental-information.

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
