# Peer review of "The implementation of rare events logistic regression to predict the distribution of mesophotic hard corals across the main Hawaiian Islands"

_PeerJ, doi:10.7717/peerj.2189_

## Round 0.1 · original submission · Major Revisions

I apologize for the delay in returning this to you; it took some time to find the correct reviewers. I have now received comments from all three reviewers, who have all offered helpful comments.

Both reviewers 2 and 3 have noted, in one way or another, the need for some rewriting; to put this study in context with previous research (reviewer 2) and to, for lack of a better term, add a bit of 'background biology' to the paper (reviewer 3).

All reviewers had no large concerns with the statistical analyses, and I consider all three sets of comments to be helpful in improving your work. Thus, my decision is 'major revision', although based on how you respond to these comments, it may become/feel more like a 'minor revision'.

·

Basic reporting

This work meets all basic reporting criteria.

Experimental design

This work meets all experimental design criteria.

Validity of the findings

This work meets all validity of findings criteria.

Additional comments

This is an impressive manuscript: well-written with clear goals, a very rich data set, rigorous statistics, and reasonable interpretations. Though I have rarely used logistic regression myself, the authors did an excellent job providing an overview of the approach, and I found it easy to follow along with the logic of each analysis (if not the specific mathematics, but others can dig into the R code if they so desire). Given the measured description of error sources and how they would affect the reliability of coral presence/absence predictions in different locations, I am satisfied that the authors have done their due diligence, and feel the manuscript is acceptable for publication. I have only a few minor suggestions.

Line 88: Please clarify the number of dives that took place at each location; e.g. "south O'ahu (X dives) and southeast Kaua'i (X dives)"

Line 91: Please provide one or two sentences describing the random five point overlay method in more detail. Does this mean for a given image, CPCe overlayed five random points, each of which were assessed for the presence of the target coral species, and if found in at least one point, the entire image was scored as "species present," and if not found at any point, as "species absent?"

Line 270: Does MHI = Main Hawaiian Islands? If so, please provide the acronym at first mention in the introduction.

Line 445: Delete "between"

Finally, it may be a consequence of the upload, but there were a few sentences where spaces appeared to be missing between words, and there may be more that I missed: Line 228 (verified the), Line 336 (separate mesophotic), Line 351 (present between), Line 372 (build a pan-Hawai'i), Line 459 (Montipora thrives), Line 460 (Leptoseris tends).

Reviewer 2 ·

Basic reporting

I think this fits all of the basic reporting criteria. My only real criticism is that there is not really sufficient background to demonstrate how these results fit in the broader field of knowledge (see my comment below about a priori hypotheses).

Experimental design

See my comment below about sampling design. I feel more details are needed to provide readers of where and when surveys were carried out.

Validity of the findings

I am not really in a position to judge the model, so will leave that to a reviewer with expertise in this area.

Additional comments

The aim of the study by Veazey et al is to develop predictive habitat suitability models for two coral genera in the mesophotic zone of Hawai’i. Studies of corals in the mesophotic have become very popular recently because of their potential role as refugia and because of improvements in remote vehicle technology that have made them more accessible. This study area is, therefore, of interest to a wide range of readers. I am not an expert in these modelling techniques, so will leave critique of that to reviewers who have expertise in this area. I guess my only real negative comment about this is that there is really no attempt a priori at developing hypotheses or potential models to explain distribution. It would be nice to see some a priori predictions based on our knowledge of coral biology from shallow water systems. I realise that this would need to be done retrospectively now that you have the results of your model, but I can’t help but feel it would make this a more interesting and readable paper.

Abstract: My first comment is that from reading the abstract, it wasn’t clear to me what the aim of the study was. I think you need a sentence defining predictive habitat suitability models and what you mean by “robust ecological assessment”.

Line 84: Should it be Hawai’ian (with an apostrophe)?

Lines 83-88: It would be good to know a bit more about the spatial arrangement of sampling dives. Is it possible that the same sites were visited more than once or any likelihood of spatial correlation? I see that you discuss this much later on (Line 404-407), but would be good to have something right at the beginning to discuss the potential problems with the spatial sampling.

Line 103: It would be good to list here which environmental covariates were used for the analysis.

Lines 283-285: These two sentences do not have any context, what do these values mean?

Lines 287-290: Suggest deleting.

Line 336: space between separate and mesophotic.

Line 350: space between offs and from.

·

Basic reporting

No comments

Experimental design

No comments

Validity of the findings

No comments

Additional comments

Dear Veazey and co-authors, I was pleased to read you article submitted to PeerJ. I think this is an exceptional attempt to explore new factors that can control the distribution of mesophotic corals in areas of potential habitat. I Am not fully qualified to comment on the statistical approach, but the rationale and procedure appear valid.

One minor criticism of the manuscript style is that it is very rooted in the statistical approach. I don't think anything needs to change there, but I would like to see some additional focus on the potential drivers of the coral distributions. I think a little more focus on the biology will make the paper more digestible to a coral reef science audience and improve its appeal. For instance, I could not find the influence of any of the individual variables or their interactions on the probability of presence/absence of the genera, aside from what is mentioned briefly in the Discussion. Perhaps this approach does not provide that type of information. If so, then please describe that in the manuscript. If there is the possibility to extract this information, then I would like to see it brought in as something we can see. Clearly this omission is not obfuscation, as you have provided all the data and code. It might also be interesting to see a figure of the probability of presence/absence with depth for each of the genera, but I leave that to you.

I will upload my marked pdf for your revision.

---

## Round 0.2 · accepted · Accept

The reviewer who examined your newly submitted version found it to be extremely well revised. I have also gone over the newest version, and found only some very small mistakes (see attached pdf) for editing, and for these reasons, my decision is 'accept'. I look forward to seeing this manuscript published.

Please ensure the minor edits in the attached pdf are taken care of in the proof version or earlier.

Reviewer 2 ·

Basic reporting

No Comments

Experimental design

No Comments

Validity of the findings

No Comments

Additional comments

The authors have done an excellent job of addressing all of the reviewer's comments. I don't have any problems recommending this manuscript for publication.